# Spatial-Temporal Distribution Pattern and Tourism Utilization Potential of Intangible Cultural Heritage Resources in the Yellow River Basin

**Bianrong Chang [1], Xinjun Ding [1],*, Jianchao Xi [2], Ruiying Zhang [1,2],* and Xianhong Lv [1,3]**

[1] College of Humanities, Tianjin Agricultural University, Tianjin 300384, China
[2] Institute of Geographic Sciences and Natural Resources Research, Chinese Academy of Sciences (CAS), Beijing 100101, China
[3] Faculty of Applied Economics, University of Chinese Academy of Social Sciences (UCASS), Beijing 102488, China
* Correspondence: jackding@pku.edu.cn (X.D.); zry8063@tjau.edu.cn (R.Z.)

**Abstract:** In this study, the spatial-temporal distribution pattern and tourism utilization potential of national ICH are analyzed based on GIS technology in the Yellow River Basin. The results show that: (1) The national ICH of the Yellow River Basin is distributed in an aggregated spatial pattern with the characteristic of "one belt and two cores". The cities in Shanxi, Henan and Shandong provinces constitute the high-density and sub-high-density continuous belts. Haidong and Chengdu are the high-density and sub-high-density core areas, respectively. (2) The five batches of national ICH present a temporal distribution pattern of "Northeast to Southwest". The distribution scope of national ICH in medium and high-density areas expands gradually and finally tends to be balanced. (3) For the single-factor evaluation of tourism utilization potential, there is an obvious gap in the evaluation grade among the cities. However, for the comprehensive evaluation of tourism utilization potential, there are 72 cities with an evaluation grade of three or above, accounting for 63%. This indicates that the tourism utilization potential of the national ICH in this basin is great. (4) The national ICH tourism in this basin can be developed from two aspects: regional differential development and regional linkage development.

**Keywords:** Yellow River Basin; intangible cultural heritage; spatial-temporal distribution; tourism utilization potential; GIS

## 1. Introduction

In 2003, the Convention for the Safeguarding of the Intangible Cultural Heritage was adopted by United Nations Education Scientific and Cultural Organization (UNESCO) [1]. Since then, the definition and content of intangible cultural heritage (ICH) were formally determined and widely popularized. The importance and urgency of protecting and inheriting ICH were thus widely recognized by many countries and regions around the world. The intangible cultural heritage, which consists of all immaterial manifestations of culture, is not only the representative of the living culture of human communities but also the carrier of cultural diversity [2]. Due to the diversity of regional culture, intangible cultural heritage has become a new engine to drive the growth of the tourism economy [3]. At present, there are some disputes on the commercialization of ICH tourism in academic circles [4–6]. In fact, if a protection mechanism to mobilize the protection power of stakeholders is built and the authenticity, integrity and inheritance of ICH are adhered to, a large number of successful practices can be carried out to promote the living heritage protection and heritage value inheritance driven by economic values and the realization of cultural and social values [7–10]. Tangible utilization of ICH can effectively enhance its attraction to tourists, on the contrary, it can actively promote the inheritance and dissemination of traditional culture.

Intangible cultural heritage is an important part of China's excellent traditional culture. China has a large number of ICHs, which are composed of different cultures of different regions and nationalities [11]. According to China Intangible Cultural Heritage Network, China has 1557 national ICH representative projects by 2021 [12], attracting extensive interest from domestic and foreign tourists [13]. The Chinese government has always attached great importance to the protection of ICH. In the past two decades, China has rolled out a series of measures to strengthen the protection of the ICH [14]. In order to promote the integrated development of cultural industry and tourism, the Ministry of Culture and the Tourism Administration of China were merged into the Ministry of Culture and Tourism in 2018, providing a feasible path for the development of ICH tourism in the new era. In recent years, great progress has been made in the integrated development of ICH and tourism in China. Many regions with rich cultural resources have been lifted out of poverty by relying on the tourism industry and achieved regional revitalization [10].

As the mother river, the Yellow River has witnessed the history of China for thousands of years and is full of rich historical heritage. The national ICH of the nine provinces in the Yellow River Basin accounts for about 30% of the total number of the country [12]. These national ICHs bear the unique cultural symbols of various regions and fully reflect cultural diversity. Over the years, most of the national ICHs in the Yellow River Basin still focus on protection and inheritance, but the ability to transform itself into a real value is too low [15]. This is mainly reflected in the fact that the number of national scenic spots built with ICH as the core is relatively small, and other industries except the tourism industry are also less involved in ICH. Against this background, in August 2021, China issued the Opinions on Further Strengthening the Protection of Intangible Cultural Heritage to increase the inheritance and utilization of the rich ICH resources in the Yellow River Basin [16]. In October 2021, China issued the Outline of Ecological Protection and High Quality Development Plan for the Yellow River Basin, which proposes to build a Yellow River cultural tourism belt with international influence and promote the integrated development of culture and tourism [17]. Taking "protection first, rescue first, rational utilization, inheritance and development" as the working principle, the national ICH of the Yellow River Basin has gradually been systematically protected, and the tourism development of the whole region has been rapidly promoted.

Under the background of China's vigorous promotion of the construction of the Yellow River National Cultural Park, the scientific protection and adaptive tourism utilization of ICH as an important part of cultural heritage has become an important practical issue. In such a linear cultural heritage area, how to reasonably identify the hot spots of ICH? How to distinguish the tourism utilization potential of ICH resources? The scientific answer to this research question has outstanding value for formulating the strategy of ICH protection and utilization of the Yellow River Cultural Park, and also has reference significance for ICH tourism protection and utilization in linear cultural heritage areas. Accordingly, the purpose of this study is to explore a research path framework that organically integrates the identification method based on the analysis of the spatial-temporal distribution pattern of the national ICH and the construction of the evaluation system for the tourism utilization potential of ICH resources, and then proposes the adaptive strategies for the tourism utilization of ICH resources in different regions. Therefore, we first analyzed the temporal and spatial distribution patterns of the national ICH in the Yellow River Basin. On this basis, we further evaluated the tourism utilization potential of national ICH resources in this region and proposed targeted tourism development strategies.

## 2. Literature Review

### 2.1. Concept of ICH

The concept of ICH first originated in Japan and was initially called "intangible cultural property" in the Law for Protection of Cultural Property promulgated in the 1950s [18]. From the 1970s to the 1990s, UNESCO referred to the concept and protection experience of the Cultural Property Protection Law, and successively put forward the concepts of

"folklore", "non-physical heritage" and "cultural tradition and folklore", etc. In 1972, UNESCO approved the Convention Concerning the Protection of the World Cultural and Natural Heritage, which mainly includes the definition of "cultural heritage" and "natural heritage" [19]. The "cultural heritage" mainly refers to monuments, groups of buildings, and sites, and emphasizes that this physical cultural heritage must have outstanding universal value from the point of view of history, art or science. In 1989, UNESCO adopted the Recommendation on the Safeguarding of Traditional Culture and Folklore [20]. Folklore (or traditional and popular culture) is the totality of tradition-based creations of a cultural community, and its connotation is basically consistent with the concept of ICH. In 1998, UNESCO started a project for the proclamation of Masterpieces of the Oral and Intangible Heritage of Humanity [21] and published the first batch of Masterpieces of the Oral and Intangible Heritage of Humanity in 2001 [22].

In 2003, UNESCO discussed and adopted the Convention for the Safeguarding of the Intangible Cultural Heritage, which clearly defined the concept of "intangible cultural heritage" from the perspective of international norms for the first time [1]. The "intangible cultural heritage" refers to the practices, representations, expressions, knowledge, skills and related as well as the instruments, objects, artifacts and cultural spaces associated therewith that communities, groups and, in some cases, individuals recognize as part of their cultural heritage. Since then, the name and concept of "intangible cultural heritage" have been officially determined, and countries around the world have set off a wave of research on intangible cultural heritage.

China has officially acceded to the Convention for the Safeguarding of the Intangible Cultural Heritage in August 2004. With the convention as a reference, the General Office of the State Council of China released the Opinions on Enhancing Preservation of Intangible Cultural Heritage in 2005 [23]. The "intangible cultural heritage" in the opinion refers to various forms of traditional cultural expression (such as folk activities, performing arts, traditional knowledge and skills, as well as related appliances, physical objects, handmade products, etc.) and cultural space that are related to the people of all nationalities and closely related to life. In 2011, the Law of the People's Republic of China on Intangible Cultural Heritage went into effect [24]. This heritage law further defines the concept of intangible cultural heritage, which refers to the forms of expression of various traditional cultures that people of all ethnic groups have inherited from generation to generation and that are considered as elements of their cultural heritage, as well as the objects and places associated with traditional cultural expressions.

### 2.2. ICH Tourism

ICH and tourism are both important products of people's cultural life, and the relationship between them is inseparable. Intangible cultural heritage is a high-quality tourism resource, while tourism is an important resource and carrier of protection and inheritance of ICH. The integration and development between ICH and tourism have a deep foundation and broad prospects. By promoting the integrated development of ICH and tourism, on the one hand, it can inject more attractive cultural content into tourism and promote better development of tourism. On the other hand, it can further improve the visibility and influence of ICH in tourism and promote the spread and rational use of ICH.

The integration of cultural relics with tourism development has currently become the world trend to drive national or regional development [25]. Intangible cultural heritage itself can be seen as a cultural event, a tool for developing sustainable tourism in rural areas and a means of positioning and marketing a destination [26,27]. Esfehani and Albrecht conduct a field survey of Qeshm Geopark in southern Iran and found ICH is of great significance to the development of tourism in places where the natural environment has strong cultural significance to local communities [28]. Through quantitative research, Giudici et al. find Sardinia's rich ICH provides an additional opportunity to increase the development level of tourism in the off-season and further promotes the sustainable development of the island's tourism [29]. At present, scholars have basically reached a

consensus on the protection of ICH, although its protection and inheritance are facing many challenges [30,31]. Wang and Bramwell believe that Government interventions should be important for determining priorities between heritage protection and tourism-related development at heritage sites [32]. Deacon suggests that the protection of ICH should make the conservation management plan, improve the system, and implement the same conservation mode as the material heritage, which requires support from all sectors [33]. The Convention for the Safeguarding of Intangible Cultural Heritage is implemented at the international and domestic levels, but the convention cannot guarantee the cross-border protection of ICH. Some scholars propose that the state parties should adopt collective intellectual property rights, especially geographical indications, to protect ICH [34]. Cheng and Yuan believe that among all measures to protect ICH, intellectual property tools and increased investment are reasonable and effective measures [35]. Cominelli and Greffe explore various approaches to the safeguarding of this heritage and point out that we should rethink and formulate cultural policies [36].

The current dilemma of ICH is its fragility and scarcity, as well as its difficulty in constructing and expressing it in a spatial-temporal dimension. Overdeveloped tourism and overcrowded tourists are not conducive to the protection of fragile historical culture and relics [37]. Therefore, new technologies and protective tourism development planning can help achieve effective living inheritance. Yoshida believes that the museum display is a model to enhance the public's protection awareness, which is a potential protective tourism development way for ICH [38]. The realization of ICH value must rely on certain material carriers, among which the way of museum display is the most effective way at present. It gives consideration to both the development of ICH tourism and the enhancement of protection awareness [39]. The construction of digital scenic areas is an effective way to achieve a win-win situation, protection and inheritance of ICH can be resolved by integrating with the scenic areas [40]. Cozzani et al. create the I-Treasures platform with different services to meet the needs of different types of users in the field of ICH education, which becomes a useful tool for organizations, schools, and institutions to promote their endangered ICH [41]. The most commonly used technologies for ICH are 3D visualization, 3D modeling, augmented reality, virtual reality and motion capture systems by which different platforms are built for tourists to effectively interact with intangible culture [42,43].

### 2.3. Application of GIS in the Study of ICH

Geographic information system (GIS) technology can realize the visual expression of spatial information and has been widely used in many fields [44–47]. At present, the applications of GIS technology in ICH research are relatively few. Using GIS to study the spatial distribution of ICH and its tourism potential evaluation has become a new perspective. GIS technology can help find the rules and characteristics for the formation and evolution of ICH to further support effective protection. Xu and Pan analyze the spatial distribution and influencing factors of four batches of different types of national ICH in China based on GIS technology [48]. Based on GIS technology, Yuan et al. analyze the spatial distribution and influencing factors of different types of national ICH heritage in Hunan province of China from the perspective of prefecture-level cities [49]. Wang et al. analyze the spatial distribution and influencing factors of different types of ICH in Shaanxi province of China from the perspective of prefecture-level cities based on GIS technology [50]. Zhang et al. use GIS technology to analyze the spatial layout and influencing factors of music ICH in the Western Hunan region of China from the perspective of prefecture-level cities [51]. Meng analyzes the spatial layout and influencing factors of national and provincial ICH in Shandong province of China from the county perspective based on GIS technology [52]. Nie et al. analyze the spatial layout and influencing factors of different types of ICH in the Yellow River Basin from the provincial perspective based on GIS technology [53]. Zhang et al. analyze evolvement rules and influencing factors of

different types of ICH in different historical dynasties in the Yellow River Basin from the perspective of historical geography based on GIS technology [54].

From the perspective of the research scope, these studies mainly focus on the spatial distribution of ICH in a certain province and pay less attention to large regions. From the perspective of research objects, they mainly focus on a certain type of ICH at the provincial, prefecture-level, and county-level. From the perspective of research content, they mainly focus on the spatial distribution of ICH and its influencing factors, but less on the temporal characteristics of ICH.

### 2.4. Tourism Utilization Potential of ICH Resources

ICH resources can only be used to attract tourists in a tangible way, and their corresponding tourism utilization potential needs to be evaluated. The tourism utilization potential of ICH refers to the development, utilization and sustainable development capacity of ICH. This capacity is a potential ability of tourism resources, which is gradually accumulated through the influence and interaction of internal and external environmental factors [55]. Ma et al. believe that it is the total supply limit that can be met by various influencing factors such as regional environment, social economy and tourism resources in a certain period of time [56].

At present, there are few quantitative evaluation studies on the tourism utilization potential of ICH. Based on an expert questionnaire survey and factor analysis correction method, the index system of tourism exploitation potential of ICH is constructed in Zhangjiajie from four aspects: stakeholder factors, tourism product development factors, heritage value factors and carrying capacity factors [57]. The tourism utilization potential of the Spanish Cultural Festival is evaluated from the value of personal distribution, the estimation of economic impact and the measurement of the efficiency of management institutions, respectively [58]. The evaluation index system of the tourism development value of ICH resources in Suzhou is constructed from resource endowment, the possibility for visual display and experience, and tourism development conditions in the heritage site [59]. Barrientos et al. build an evaluation model for the competitiveness of cultural and natural heritage in rural areas [60]. Wang et al. analyze the tourism utilization potential of ICH in Shanxi province in central China based on the resource-market-product theory [61].

In general, the research on ICH mainly focuses on its connotation, authenticity, value, protection and development, as well as its impact on tourism development, while few studies are concerned with the spatial-temporal distribution of regional ICH resources and the quantitative evaluation of tourism utilization potential. In this study, we analyze the spatial-temporal characteristics of ICH in the Yellow River Basin, based on the cognition of the concept of tourism utilization potential, and establish an evaluation index system of the tourism utilization potential of ICH resources. On this basis, we further discuss the corresponding development strategies of national ICH tourism in the Yellow River Basin.

## 3. Materials and Methods

### 3.1. Study Area

The Yellow River, which originates from the Joguzongli Basin, flows through nine provinces of China and flows into the Bohai Sea in the Kenli District of Shandong province. The basin is high in the west and low in the east and has a variety of geomorphic types. The average elevation of the western region is more than 4000 m, which is composed of a series of high mountains, with perennial snow cover and glacier landforms. The elevation of the middle part of the basin is 1000–2000 m, which is a loess landform. The eastern part of the basin is mainly the Yellow River alluvial plain. The Yellow River Basin belongs to a typical monsoon climate, with large temperature differences and uneven precipitation distribution.

The Yellow River Basin has unique natural scenery, profound cultural heritage and rich ICH. However, its complex and changeable natural geographical environment and fragile ecology have a great impact on the development of the economy, society and culture

in the region. The economic development gap among provinces in the Yellow River Basin is large, and the problem of unbalanced and inadequate development is more prominent. In terms of GDP, Shandong province has the highest GDP of 70,540.48 billion yuan in 2019, while Qinghai province has the lowest GDP of 294.107 billion yuan. Except for Ningxia province, the tourism revenue of all provinces accounts for more than 15% of the GDP, among which the total tourism income of Shanxi province accounts for 47% of the GDP.

Figure 1 gives the geographical location of the Yellow River Basin. Based on statistical data from different administrative regions, we select 115 prefecture-level cities of nine provinces to study the spatial-temporal distribution pattern and tourism utilization potential of national ICH Resources in the Yellow River Basin.

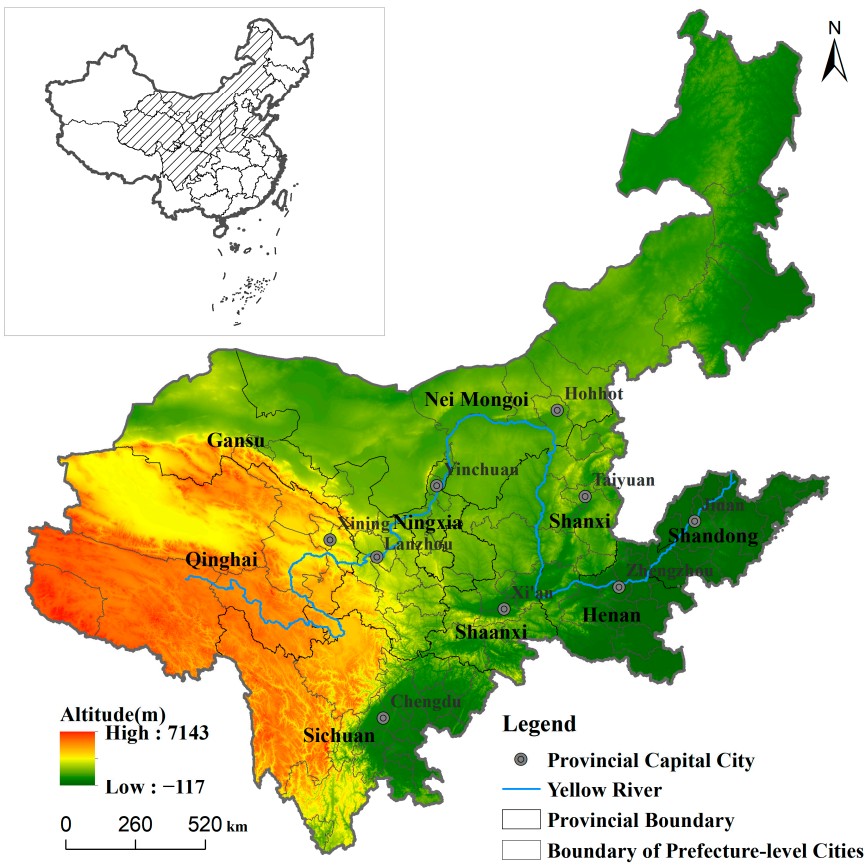

**Figure 1.** The geographical location of the Yellow River Basin.

### 3.2. Data Sources

The national ICH data at the prefecture level in the Yellow River Basin comes from the five batches of the national intangible cultural heritage lists published by the Chinese Ministry of Culture and Tourism. There are 986 items in total, which are of outstanding value nationwide. It should be noted that the national ICH project would be divided into multiple items according to its declared location, while the projects with territories all over the province have not been considered in this study. According to The First Batch of the National Intangible Cultural Heritage List and The Second Batch of National Intangible Cultural Heritage List published by China Intangible Cultural Heritage Network in 2008 [17], the national ICH resources in the Yellow River Basin can be divided into ten categories: folk literature, traditional music, traditional dance, traditional drama, Quyi, traditional sports, recreation and acrobatics, traditional art, traditional craft, traditional medicine, and folkways. The ten types of national ICH in the Yellow River Basin account for 7.9%, 13.4%, 9.0%, 13.9%, 5.6%, 4.8%, 12.8%, 15.6%, 4.2% and 12.8%, respectively. In this study, the relevant geographic coordinate information is then obtained using Baidu

Map Picking Tool and imported into ArcGIS10.8 to build a database of national ICH in the Yellow River Basin.

The data of evaluation index for tourism utilization potential are collected from China Intangible Cultural Heritage Website, Chinese Ministry of Culture and Tourism website, provincial people's government websites, Chinese Government website, National Traditional Village website, the provincial statistical yearbooks, statistical bulletins, government work reports and relevant official websites. The name of the city at the prefecture level is taken as associated fields, which are imported to ArcGIS10.8 to build the evaluation index database of the tourism utilization potential of national ICH resources in the Yellow River Basin.

### 3.3. Methods

The nearest neighbor index, kernel density analysis, standard deviation ellipse and spatial autocorrelation methods are used to analyze the spatial and temporal distribution characteristics of the national ICH in the Yellow River Basin. Based on the resource-market-product theory, the evaluation system for tourism utilization potential of national ICH is constructed from the core resources, tangible carriers, and external resources. Classified evaluation of single factor index is implemented by ArcGIS 10.8, and the entropy weight method is used to calculate the weight for each index. Finally, the comprehensive evaluation grades for tourism utilization potential of national ICH in each prefecture-level city are calculated by spatial overlay analysis.

#### 3.3.1. Nearest Neighbor Index

The nearest neighbor index is used to identify the spatial concentration and distribution of point elements. It is the spatial distribution pattern of points measured by the nearest neighbor distance [62]. The calculation formula is as follows:

$$R = \frac{\overline{r_1}}{\overline{r_E}} \tag{1}$$

where $R$ is the nearest neighbor index, $\overline{r_1} = \frac{1}{n_1} \sum_{i \in 1} r_{ij}$ is the average nearest neighbor distance, $\overline{r_E} = \frac{1}{2\sqrt{n/A}}$ is the theoretical nearest distance, $A$ is the regional area, $n$ is the number of national ICH. When $R = 1$, it indicates that the regional national ICH sites are randomly distributed. When $R > 1$, it indicates that the distribution tends to be uniform. When $R < 1$, it indicates that the distribution tends to be aggregated.

#### 3.3.2. Kernel Density Analysis

Kernel density analysis is a non-parametric method for estimating the point elements distribution intensity in the study area by calculating the input data, which reflects the radiation intensity of the core area to the surrounding area [63]. The density of the points is related to the probability of the events. The denser the point sets are, the higher probability of the events. On the contrary, the lower probability of the events. The calculation formula is as follows:

$$f(x) = \frac{1}{nh} \sum_{i=1}^{n} k\left(\frac{x - x_i}{h}\right) \tag{2}$$

where, $f(x)$ is the kernel density estimate, $k\left(\frac{x-x_i}{h}\right)$ is the kernel function, $n$ is the number of national ICH, $h > 0$ is the bandwidth, $x - x_i$ is the estimated distance from $x$ to $x_i$. The higher value of $f(x)$ indicates the denser feature distribution.

#### 3.3.3. Standard Deviation Ellipse

The standard deviation ellipse (SDE) analysis method is an algorithm for analyzing the direction and distribution of points at the same time, which can reflect the overall characteristics of the spatial distribution of elements from multiple angles [64]. It is essentially

the spatial point pattern analysis, which can make full use of the distance information between points to provide more comprehensive spatial scale information [65]. It mainly involves the center, area, rotation angle, long and short semi axis and other parameters of the ellipse. The longitude and latitude of the ellipse center indicate the relative location of the distribution of national ICH in the Yellow River Basin, and the size of the ellipse area indicates the distribution range of national ICH. The long axis indicates the main direction of the spatial distribution of national ICH, while the short axis indicates the distribution range of national ICH. The longer length of the two axes signifies more discrete spatial distribution. Furthermore, the larger difference in length between the long axis and the short axis signifies and greater ellipticity of the ellipse indicate more obvious directivity. The rotation angle reflects the change of the main trend direction of the spatial distribution of national ICH.

### 3.3.4. Spatial Autocorrelation

Spatial autocorrelation is an important indicator reflecting the degree of correlation between geographical phenomena or attribute values in a regional unit and the same of those in the adjacent regional units [66]. Global Moran's I index is used to test the correlation between two adjacent units in the study area. Getis-Ord $G_i^*$ could be used to recognize the local spatial autocorrelation feature. This study uses Moran's I index for global autocorrelation analysis, the absolute value of the index close to 1 indicates strong spatial correlation. Uses the hot spot analysis index (Getis-Ord $G_i^*$) to conduct local spatial autocorrelation analysis, the index close to 0 indicates random distribution. When the index is positive, the aggregation region is a "positive hot spot region". When the index is negative, the aggregation region is a "negative hot spot region". The calculation formula is as follows:

$$I = \frac{n \sum_{i=1}^{n} \sum_{j=1}^{n} w_{i,j}(x_i - x)(x_j - x)}{\sum_{i=1}^{n} \sum_{j=1}^{n} w_{i,j \sum_{i=1}^{n} (x_i - x)^2}}, i \neq j \tag{3}$$

$$G_i^* = \frac{\sum_{j=1}^{n} w_{i,j} x_j - x \sum_{j=1}^{n} w_{i,j}}{\sqrt{\frac{\sum_{j=1}^{n} x_j^2}{n} - x^2} \sqrt{\frac{n \sum_{j=1}^{n} w_{i,j}^2 - \left(\sum_{j=1}^{n} w_{i,j}\right)}{n-1}}} \tag{4}$$

where, $n$ refers to the number of research space units, and it refers to the number of prefecture-level cities in the Yellow River Basin in this study. $w_{i,j}$ is the spatial weight coefficient of $i$th region and $j$th region, reflecting the spatial relationship of $i$th region and $j$th region. In this study, the spatial weight file is automatically generated by the inverse distance method. $x_i$ is the number of occurrences of a phenomenon in the $i$th unit, and it is the number of national ICH in each city. $x$ is the average number of a phenomenon in all research units, and it is the average number of national intangible cultural heritages in each city in this study. Moran's I index is between $-1$ and 1.

### 3.3.5. RMP Analysis

RMP is an analysis of product (P) based on resources (R) and market analysis (M), which helps build a product-centered tourism development planning theory [67]. Before product development, it needs to analyze the tourism resource endowment, surrounding infrastructure support, potential market, economic conditions and other external environments. Based on this theory, this study analyzes the relationship between tourism resources and product and builds an index system for evaluating the tourism utilization potential of regional ICH resources.

### 3.3.6. Entropy Weight Method

The entropy weight method is an objective evaluation method that measures the value dispersion of the data itself. It is used for comprehensively scoring samples by multiple indexes to achieve the comparison between samples, and determine weight coefficient

according to difference degree to objectively show the importance of an index in the system [68]. The calculation formula is as follows:

$$H_j = -k \sum_{i=1}^{m} p_{ij} ln p_{ij}, \ k = \frac{1}{lnm} \tag{5}$$

$$p_{ij} = \frac{X'_{ij}}{\sum_{i=1}^{m} X'_{ij}} \tag{6}$$

where $H_j$ is the entropy value of the $j$th index, $p_{ij}$ is the proportion of index in $i$th city (state), $m$ is the number of cities (states) in the study area, $X_{ij}$ is the index value of $j$th item in $i$th city (state), $X'_{ij}$ is the value of $X_{ij}$ after dimensionless processing.

Then, the index weight value is determined according to the entropy value, and the calculation formula is:

$$W_j = \frac{1 - H_j}{\sum_{j=1}^{n}(1 - H_j)} \tag{7}$$

where $n$ is the number of indexes in the evaluation index system.

## 4. Results and Analysis

### 4.1. Temporal and Spatial Pattern of National ICH Resources

4.1.1. Spatial Distribution Pattern

Overall, the nearest neighbor indexes of national ICH resources in the Yellow River Basin are all less than 1, the Z-score is less than −2.58, and the $p$-value is less than 0.01 (Table 1). The results suggest that the national ICH in the basin is clustered. Among them, the second batch of national ICH has the strongest aggregation degree due to its largest number of national ICH. There are 427 items in the second batch, which account for 43.3% of the total number. However, the nearest neighbor index of national ICH has gradually increased since the second batch. It signifies that the distribution of national ICH has gradually become uniform after 2008.

**Table 1.** The nearest neighbor index (NNI) of each batch of national ICH in the Yellow River Basin.

| Batch | Quantity (Item) | Average Nearest-Neighbor Distance (km) | Expected Nearest-Neighbor Distance (km) | NNI | Z-Score | $p$-Value | Spatial Distribution Pattern |
|---|---|---|---|---|---|---|---|
| First | 174 | 44.466 | 80.801 | 0.550 | −11.348 | 0.000 | clustered |
| Second | 427 | 21.783 | 54.034 | 0.403 | −23.595 | 0.000 | clustered |
| Third | 145 | 49.995 | 82.244 | 0.608 | −9.033 | 0.000 | clustered |
| Fourth | 127 | 60.565 | 88.773 | 0.682 | −6.851 | 0.000 | clustered |
| Fifth | 113 | 79.198 | 91.392 | 0.867 | −2.713 | 0.007 | clustered |
| All | 986 | 11.368 | 36.022 | 0.316 | −41.115 | 0.000 | clustered |

For different regions in the upstream, midstream and downstream of Yellow River Basin, the nearest neighbor indexes are 0.42, 0.28 and 0.34, respectively, showing an aggregation distribution trend. As shown in Table 2, the nearest neighbor indexes of the national ICH for each province are all less than 1, which also shows a clustering distribution trend. This phenomenon is due to the significant differences in the number of national ICHs in each province. Among them, Shandong province in the downstream and Shanxi province in the midstream is rich in national ICH, accounting for 18.26% and 18.15%, respectively. However, Ningxia Hui Autonomous Region in the upstream has the least national ICH resources, accounting for only 1.93%. From the perspective of geographical location, the distribution of national ICH shows a polarization feature. Due to better geographical environment, faster economic development, and the rich and diverse cultures bred in the

process of historical development, there are more national ICH resources in the middle and lower reaches. However, due to the large distribution of deserts and mountains and the late start of economic and cultural development, there are fewer national ICHs in the western provinces of the upstream region.

**Table 2.** The nearest neighbor index of national ICHs in different watersheds and provinces.

| Province | Quantity (Item) | Average Nearest-Neighbor Distance (km) | Expected Nearest-Neighbor Distance (km) | NNI | Z-Score | *p*-Value | Spatial Distribution Pattern |
|---|---|---|---|---|---|---|---|
| Qinghai | 82 | 15.337 | 35.138 | 0.436 | −9.762 | 0.000 | clustered |
| Sichuan | 149 | 15.800 | 31.149 | 0.507 | −11.507 | 0.000 | clustered |
| Gansu | 81 | 15.041 | 43.396 | 0.347 | −11.250 | 0.000 | clustered |
| Ningxia | 19 | 12.020 | 23.453 | 0.513 | −4.065 | 0.000 | clustered |
| Upstream | 331 | 14.959 | 35.717 | 0.419 | −20.228 | 0.000 | clustered |
| Nei Monggol | 96 | 26.787 | 66.012 | 0.406 | −11.138 | 0.000 | clustered |
| Shaanxi | 79 | 12.909 | 25.358 | 0.509 | −8.348 | 0.000 | clustered |
| Shanxi | 179 | 5.045 | 15.644 | 0.323 | −17.340 | 0.000 | clustered |
| Midstream | 354 | 12.443 | 44.571 | 0.279 | −25.945 | 0.000 | clustered |
| Henan | 121 | 10.236 | 20.204 | 0.507 | −10.382 | 0.000 | clustered |
| Shandong | 180 | 4.678 | 17.786 | 0.263 | −18.915 | 0.000 | clustered |
| Downstream | 986 | 6.889 | 20.219 | 0.341 | −21.882 | 0.00 | clustered |

### 4.1.2. Spatial Distribution Density

Based on ArcGIS10.8 software, the kernel density analysis of national ICH in the Yellow River Basin is carried out on the whole and in batches, respectively, and the natural breakpoint method is then used to divide them into five categories: low-value area, sub-low-value area, medium-value area, sub-high-value area and high-value area. It can be seen from Figure 2 that the national ICH resources in the Yellow River Basin show a spatial distribution pattern of "one belt and two cores". Some cities in Shanxi, Henan and Shandong provinces along the middle and lower reaches of the Yellow River constitute a high-density and a sub-high-density continuous belt. Among them, Haidong and Chengdu are high- and sub-high-density core areas, respectively, and the sub-high-density areas are located at the periphery of the high-density regional cities. For the results of each batch:

(1) The first batch of national ICH in the Yellow River Basin has formed two high-density continuous belts; three sub-high-density continuous belts and a sub-high-density core area in space. Among them, two high density continuous belts are located in Yuncheng; Jincheng of Shanxi province; Jiaozuo; Zhengzhou; Xinxiang and Hebi of Henan province; Xining and Haidong of Qinghai province; Lanzhou and Linxia of Gansu province. The sub-high-density zone is located in the peripheral cities of the high-density zone; where Jinan; Zibo; Weifang and Qingdao in Shandong province form a sub-high-density zone. Chengdu of Sichuan province is a sub-high-density core area

(2) The second batch of national ICH in the Yellow River Basin has formed a high-density continuous belt, a high-density core area, a sub-high-density continuous belt and three sub-high-density core areas in space. Compared with the results of first batch, the high-density continuous zone has been significantly expanded. Cities along the line "Taiyuan-Zhengzhou-Jinan" form a "$\sqrt{}$" shape. Chengdu has developed into a high-density core area. The sub-high-density core area is located in Xining and Haidong of Qinghai province, Qingdao of Shandong province, Meishan and Deyang of Sichuan province, which are also peripheral cities in the high-density zone.

(3) The third batch of national ICH in the Yellow River Basin has formed a high-density core area, two sub-high-density continuous belts and one sub-high-density core

area in space. The high-density core area is located at the junction of Linfen, Jincheng and Yuncheng in Shanxi province, and its scope is significantly reduced. One sub-high-density continuous belt is located in the periphery of the high-density core area, mainly in the south of Xinzhou of Shanxi province, the north of Zhengzhou of Henan province, and Heze of Shandong province. The other sub-high-density continuous belt is centered in Weifang of Shandong province, extending to the surrounding cities of Zibo, Binzhou, Dongying, and Qingdao.

(4) The fourth batch of national ICH in the Yellow River Basin has formed four high-density core areas, a sub-high-density continuous belt and four sub-high-density core areas in space. The high-density area is still mainly located in some cities in Shanxi, Henan and Shandong provinces. Liangshan Prefecture in Sichuan province has become a new high-density core area. The sub-high-density continuous zone and sub-high-density core area are still located at the periphery of the high-density core area, at the junction of Haidong, Lanzhou and Linxia, and at the junction of Pingliang, Guyuan, Tianshui and Dingxi.

(5) The fifth batch of national ICH in the Yellow River Basin has formed seven high-density core areas in space, and the sub-high-density areas are located in the periphery of the high-density core city. Compared with the previous batches, the distribution of medium and high-density areas is more balanced. The surrounding urban area centered on Yinchuan city of Ningxia Hui Autonomous Region has become a new high-density core area. Influenced by the policy of the Outline of Ecological Protection and High Quality Development Plan for the Yellow River Basin, Ningxia Hui Autonomous Region has paid more attention to the inheritance and protection of local culture. Therefore, the number of national ICH has increased significantly in recent years.

In general, the spatial distribution of national ICH resources in the Yellow River Basin is significantly different, and the high-value areas are still mainly distributed in the middle and lower reaches of the basin. As time goes on, the distribution range of the middle and high-value areas gradually expands, and the distribution areas become more dispersed and tend to be balanced. In comparison, the geographical environment of the middle and lower reaches of the Yellow River Basin is more suitable for human survival, thus creating a rich and colorful historical civilization. In comparison, the geographical environment of the middle and lower reaches of the Yellow River Basin is more suitable for human survival, thus creating a rich and colorful historical civilization. On the contrary, the environment in the upper reaches of the basin is worse and the traffic is more blocked, resulting in less communication with the outside world. Most of the cultural heritage currently reserved in the upstream area is unique to ethnic minorities, which makes it difficult to excavate and define the local ICH.

### 4.1.3. Spatial Differentiation Characteristics

Figure 3 shows the spatial distribution changes of five batches of national ICH resources in the Yellow River Basin from 2006 to 2021. It can be seen from Figure 3 that the national ICH of the Yellow River Basin moved from northeast to southwest during this period. The long semi axis of the standard deviation ellipse is always larger than the short semi axis, the rotation angle has increased, and the ellipse area has increased slightly, which indicates that the national ICH in the Yellow River Basin shows an outward diffusion trend, and the distribution gradually tends to be balanced. It is worth noting that the standard deviation ellipse of the third batch of national ICH has the lowest area, long and short semi axis values, and has the biggest changes. The reason is that the number of the second batch of national ICH projects increased significantly in 2008, reaching 427. However, national ICH projects appeared the phenomenon of emphasizing declaration but neglecting protection in the later period. In view of this, China began to implement the Law of the People's Republic of China on Intangible Cultural Heritage in 2011, and strictly strengthened the approval, supervision and inspection of national ICH projects. By controlling the declaration process, the number of national ICH projects approved in the subsequent batches decreased significantly.

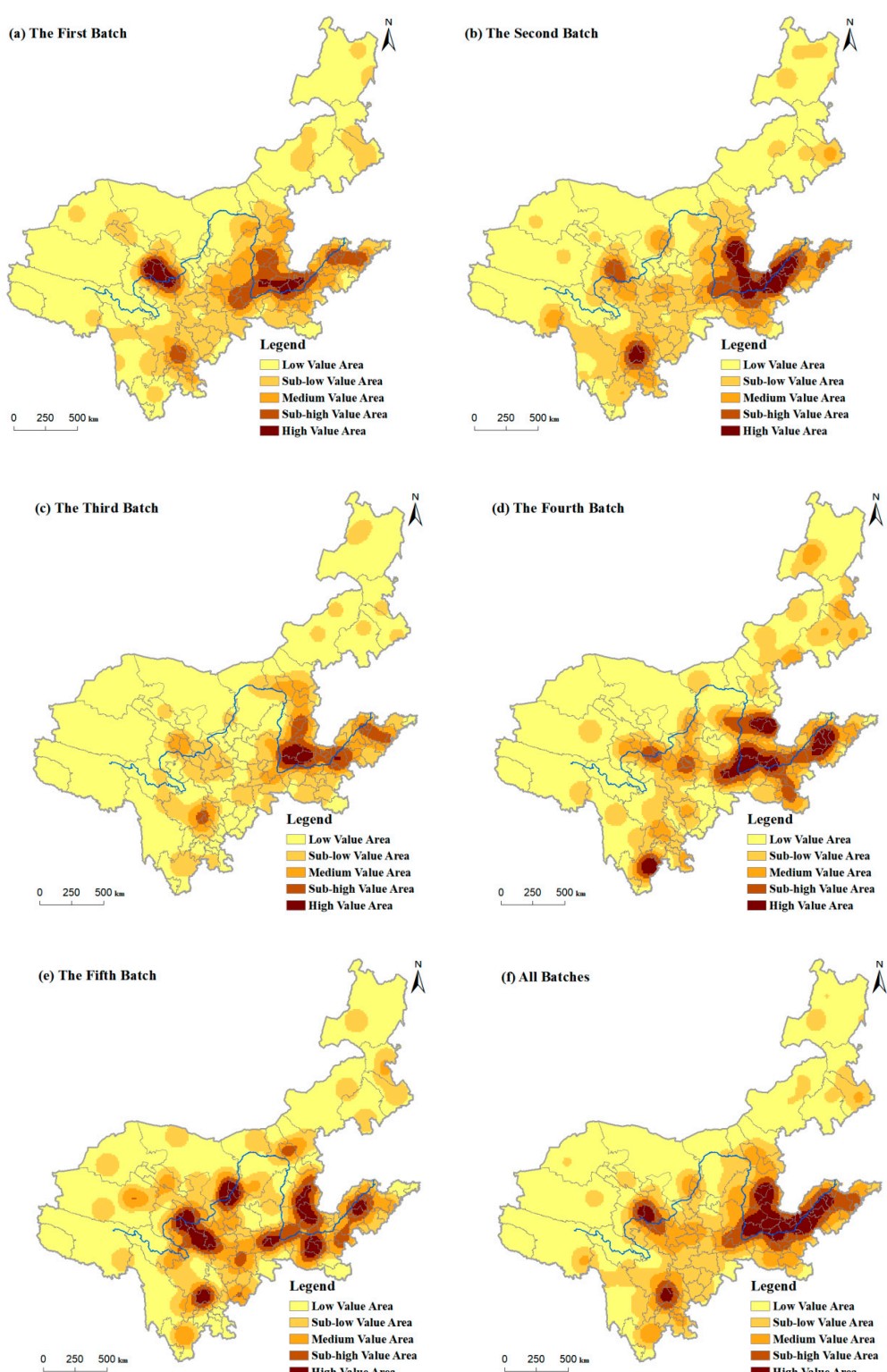

**Figure 2.** The kernel density analysis of national ICH in the Yellow River Basin.

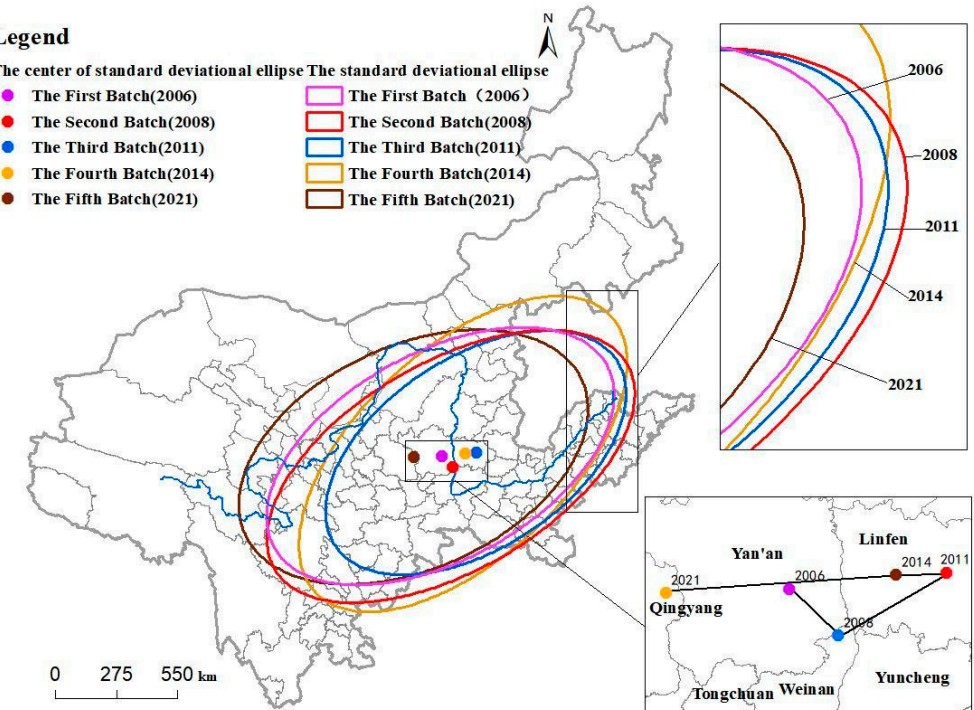

**Figure 3.** The standard deviation ellipse of national ICH and its central locus in the Yellow River Basin.

Taking the third batch in 2011 as the dividing point, the moving track of the gravity centers of the standard deviation ellipse can be divided into two stages, 2006–2011 and 2011–2021. In 2006, the spatial distribution center of the first batch was located in Yan'an city of Shaanxi province and then moved eastward and the second and third batches of national ICH were located in Weinan city of Shaanxi province and Linfen city of Shanxi province, respectively. During this period, the proportion of national ICH in Shandong province and Shanxi province continued to increase in the second and third batch, accounting for 21.78% and 15.69%, 22.76% and 29.66%, respectively, which caused the spatial distribution center to move eastward. After 2011, the spatial distribution centers for the fourth and fifth batches of national ICH began to move westward. For the fourth batch of national ICH declared in 2014, the proportion of national ICH for Shandong province and Shanxi province in the east decreased significantly, while that of Inner Mongolia and Sichuan province in the west increased significantly. For the fifth batch of national ICH declared in 2021, the proportion of national ICH for Shanxi province and Shandong province continued to decrease, while that for Gansu province, Ningxia Province and Qinghai province in the upper reaches of the Yellow River Basin increased significantly. During this period, the spatial distribution center moved westward obviously, and was finally located in Qingyang city of Gansu province. Benefiting from the implementation of the Outline of Ecological Protection and High Quality Development Plan for the Yellow River Basin, the distribution of national ICH in each province tends to be balanced, which makes the national ICH in the basin inherited and protected in all aspects.

### 4.1.4. Spatial Autocorrelation Characteristics

The global autocorrelation analysis shows that the Moran's I value of national ICH in the Yellow River Basin is 0.391, which is greater than the expected index. The *p*-value is less than 0.01, and the z-score is positive. The results indicate that there is obvious spatial autocorrelation in the spatial distribution of national ICH in the overall scope, that is, there is a spatial aggregation (Table 3). For the first to the fourth batch of national ICH in the Yellow River Basin, there is also a phenomenon of aggregation in its spatial distribution.

The Moran's I value of the fifth batch is −0.009, which is smaller than the expected index and has no statistical significance. It is distributed randomly in the whole region.

**Table 3.** The results of global autocorrelation analysis of national ICH.

| Batch | Moran's I | Expected Index | Variance | Z-Score | *p*-Value | Result |
|--------|-----------|----------------|----------|---------|-----------|-----------|
| All | 0.391 | −0.009 | 0.001 | 12.574 | 0.000 | clustered |
| First | 0.134 | −0.009 | 0.001 | 4.598 | 0.000 | clustered |
| Second | 0.300 | −0.009 | 0.001 | 9.793 | 0.000 | clustered |
| Third | 0.218 | −0.009 | 0.009 | 7.561 | 0.000 | clustered |
| Fourth | 0.116 | −0.009 | 0.001 | 4.012 | 0.000 | clustered |
| Fifth | −0.009 | −0.009 | 0.001 | −0.004 | 0.997 | random |

As shown in Figure 4, the hot spot analysis demonstrates that the hot spots of the first batch of national ICH are located in 40 cities, mainly distributed in the central and southern Shanxi province, Henan province, the central and western Shandong province, Haibei in Qinghai Province and Hainan Tibetan Autonomous Prefecture. The cold spots are located in five cities, including Wuhai, Alxa League, Bayannur, Shizuishan, and Yinchuan. The hot spots of the second batch of national ICH are located in 41 cities, mainly in the south-central Shanxi, Henan and Shandong provinces. The cold spots have increased significantly to 18 cities, and its distribution scope expanded to the south, adding some cities in Gansu and Shaanxi provinces. Although the hot spots of the third batch of national ICH are still located in 40 cities, the scope of the eastern Shandong province has narrowed, on the contrary, the scope of the northern Shanxi province has increased. The distribution range of the cold spots moves southward, covering 22 cities in total, which are mainly located in most cities in Sichuan province and neighboring Shaanxi province. The hot spots of the fourth batch of national ICHs have reached 40 cities in total, of which the number of cities in the north of Shanxi province has decreased, while Shaanxi province in the west has added Yan'an, Tongchuan and Weinan cities. However, the coverage of the cold spots has been greatly reduced to eight cities, mainly located in the surrounding areas centering on Deyang city of Sichuan province. Compared with the results of previous batches, the hot spots of the fifth batch of national ICH were significantly reduced. It has 14 cities in total, including 9 cities along the Yellow River in Henan province, Changzhi and Jincheng in Shanxi province, Heze and Jining in Shandong province and Yulin in Shaanxi province and there is no cold spots area in this batch. From all five batches of national ICH, the hot spots are concentrated in most cities in the central and southern parts of Shanxi, Henan and Shandong provinces, while the cold spots are mainly concentrated in some cities in Inner Mongolia, Ningxia and Sichuan provinces.

On the whole, the hot spots of national ICH in the Yellow River Basin are always located in most cities in central and southern Shanxi, Henan, and Shandong provinces. This is mainly because the provinces in the middle and lower reaches of the Yellow River Basin are relatively economically developed and densely populated, while the development of Inner Mongolia, Ningxia, Sichuan and other provinces in the upstream region is relatively backward.

*4.2. Evaluation of Tourism Utilization Potential of National ICH Resources*

4.2.1. Index System for Evaluation of Tourism Utilization Potential of ICH

According to RMP theory, tourism resources can be divided into core resources and supporting resources [67]. The core resources mainly refer to tourist attractions, while the supporting resources mainly refer to regional tourism supporting facilities. The transformation relationship between resources and products can be divided into three types: ① The core resources are very high quality, which can be easily transformed into superior products (such as Terra Cotta Warriors in Xi'an city). ② The core resources are of low grade and require huge investment to upgrade to high-quality products. ③ The core resource is intangible culture, which needs to be combined with tangible physical resources to form

products, such as red tourism composed of red revolutionary spirit and revolutionary base areas. Intangible cultural heritage is an intangible culture, and its development needs tangible carriers. Therefore, we evaluate the tourism utilization potential of ICH resources in three aspects: core resources, tangible carriers and the external environment.

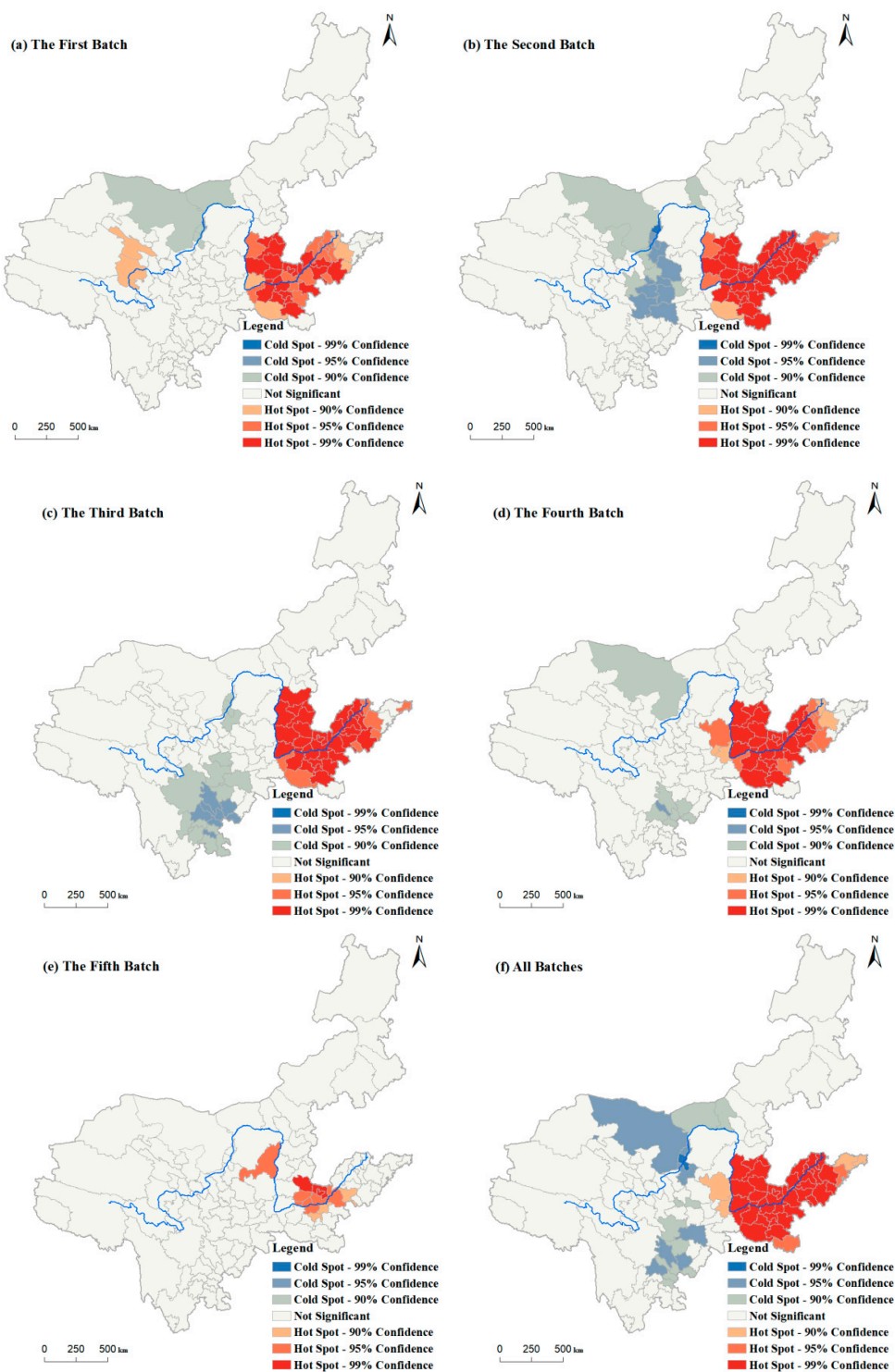

**Figure 4.** The results of hot spot analysis ($\alpha = 0.1$).

The core resources are mainly positively related to the number and category of ICH, category advantage in the region [67]. ICH projects can only have tourism attraction if they are clustered and distributed in a certain number. Considering the participation

and experience of tourism, the category advantages of ICH are mainly represented by handicrafts, folk festivals and performances. The tangible carrier mainly considers the number of traditional settlements, historical and cultural cities, towns and villages and the advantages of national 4A and 5A scenic spots. Traditional settlements, historical and cultural cities, towns and villages have greatly preserved the authenticity of intangible culture and restored the features of ICH in daily life. The scenic spot is closely related to the utilization of ICH resources, which will be combined with the local intangible culture during development. The external environment mainly considers three indexes: regional economic base, tourism development level and traffic conditions. As shown in Table 4, a total of 8 indexes are selected to establish an evaluation system for the tourism utilization potential of national ICH resources in the Yellow River Basin, and the entropy weight method is used to calculate the weight of each index.

**Table 4.** Evaluation index system of tourism utilization potential of national ICH.

| Evaluating Aspects | Specific Index | Explanation of the Index | Weight |
|---|---|---|---|
| Core resources | Q1: Number of national ICH | The total number of national ICH in each city | 0.118 |
| | Q2: Category of national ICH | The total number of category of national ICH in each city | 0.063 |
| | Q3: Category advantage of national ICH | Q3 = D/2 + 3D/365 + E/2. C, D, and E is the total number of handicrafts, folkways, performances (Quyi, traditional dance, traditional drama and traditional music), respectively. | 0.142 |
| Tangible carrier | Q4: Advantages of scenic spots | Q4 = 5N1 + 4N2. N1 and N2 is the total number of national 5A scenic spots and 4A scenic spots in each city, respectively. | 0.079 |
| | Q5: Number of traditional settlements | Q5 = M1 + M2 + M3. M1, M2, and M3 is the total number of national famous historical and cultural towns, national famous historical and cultural villages, national traditional villages, respectively. | 0.259 |
| External environment | Q6: Regional economic development | GDP per capita (unit: yuan) | 0.091 |
| | Q7: Tourism development level | Tourism income (unit: million yuan) | 0.168 |
| | Q8: Traffic conditions | Q8 = 5T1 + 4T2 + 3T3 + 2T4 + T5. T1, T2, T3, T4, and T5 is the total number of air stations, railway stations, expressway, national highways, and provincial roads. | 0.080 |

### 4.2.2. Grading of Single Factor Evaluation

Based on ArcGIS10.8, the grading standard of single-factor evaluation of tourism utilization potential is given in the Yellow River Basin (Table 5). Using the natural breakpoint splitting method, the evaluation of the specific index for the evaluation unit is divided into five grades.

**Table 5.** The grading standard of single-factor evaluation.

| Grade | Number of National ICH | Category of National ICH | Category Advantages | Advantages of Scenic Spots | Number of Traditional Settlements | Regional Economic Development | Tourism Development Level | Traffic Conditions |
|---|---|---|---|---|---|---|---|---|
| 1 | 0–4 | 0–1 | 0–1.02 | 0–25 | 0–8 | 14,256–28,951 | 2.81–251.38 | 29–47 |
| 2 | 5–9 | 2–3 | 1.02–2.52 | 25–45 | 8–22 | 28,951–46,168 | 251.38–563 | 47–93 |
| 3 | 10–14 | 4–5 | 2.52–4.05 | 45–74 | 22–42 | 46,168–66,313 | 563–936.6 | 93–134 |
| 4 | 15–20 | 6–7 | 4.05–6.02 | 74–125 | 42–96 | 66,313–97,564.3 | 936.6–1955.9 | 134–224 |
| 5 | 21–32 | 8–10 | 6.02–10.50 | 125–197 | 96–202 | 97,564.3–173,069 | 1955.9–4650 | 224–371 |

(1)　Core resources

Figure 5 gives the evaluation grade of the three specific indexes of core resources. It can be seen that the total number of national ICHs in cities of Yellow River Basin is large and the types are relatively diverse. However, the spatial distribution is uneven, and the category advantage is not high. ① In terms of the number of national ICH, there are relatively few regions with high grade evaluation (Grade 4 and above), including 21 prefecture-level cities, accounting for 18% of the total number of evaluation units. They are mainly Heze city in Shandong province, Jinzhong, Changzhi, Linfen, Jincheng in Shanxi province, Western Minority Autonomous Prefecture and Chengdu city in Sichuan province, and Haidong city in Qinghai province, among which Heze city has the highest level with 32 items. A total of 67% of prefecture-level cities in the evaluation unit have low grade evaluation (Grade 2 and below), and 12 prefecture-level cities have only one or no national ICH. ② In terms of the number of categories of national ICH, the prefecture-level cities with high grade evaluation (Grade 4 and above) account for 41% of the total number. Among them, the diversity is most prominent in the area from the south of Haidong in Qinghai province to Chengdu in Sichuan province, from the south of Taiyuan in Shanxi to Weinan in Shaanxi, and along Jining-Heze-Tai'an in Shandong province. ③ From the category advantage of national ICH, they are not high on the whole. There are 20 prefecture-level cities with high grade evaluation (Grade 4 and above), accounting for only 17% of the total number. Among them, Heze in Shandong province, Changzhi, Linfen and Yuncheng in Shanxi province, and Chengdu, Aba Tibetan Autonomous Prefecture and Ganzi Tibetan and Qiang Autonomous Prefecture in Sichuan province have the highest category advantage (Grade 5). It shows that these regions have obvious advantages in traditional craft, folkways, Quyi, traditional dance, traditional drama, traditional music and other national ICH resources, and have strong tourism attraction.

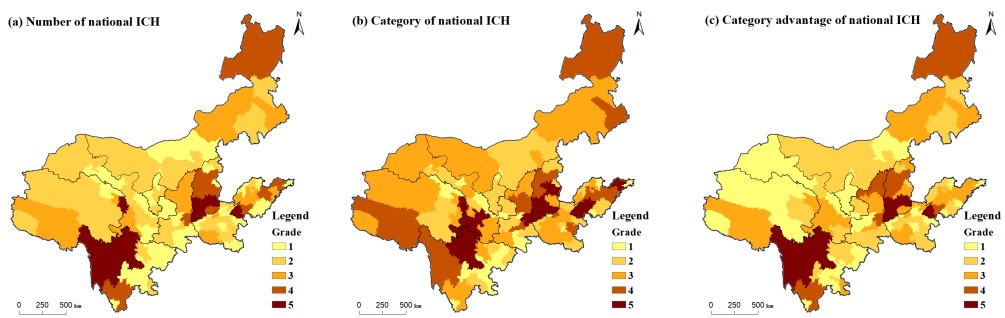

**Figure 5.** The evaluation grade of specific indexes of tangible carrier.

(2)　Tangible carrier

Figure 6 gives the evaluation grade of the two specific indexes of tangible carrier. It can be seen that the tangible carrier of national ICH in the Yellow River Basin has general advantages, which have a certain impact on the evaluation of national ICH utilization potential. ① From the advantages of scenic spots, there are relatively few areas with high grade evaluation (Grade 4 and above). There are 19 prefecture-level cities, accounting for 16% of the evaluation unit. Most of them are located in the middle and lower reaches of the Yellow River. Among them, Chengdu has the highest level, with a total of 49 national 4A and 5A scenic spots. ② In terms of the number of traditional settlements, there are fewer areas with high grade evaluation (Grade 4 and above). There are only 8 prefecture-level cities, including Jincheng, Linfen, Changzhi, Jinzhong, Yangquan and Luliang in Shanxi province, Haidong in Qinghai province and Ganzi Tibetan Autonomous Prefecture in Sichuan province, accounting for 7% of the evaluation unit. Among them, Jincheng in Shanxi province has the highest level, with 5 nationally famous historical and cultural towns, 35 national famous historical and cultural villages and 182 national traditional villages.

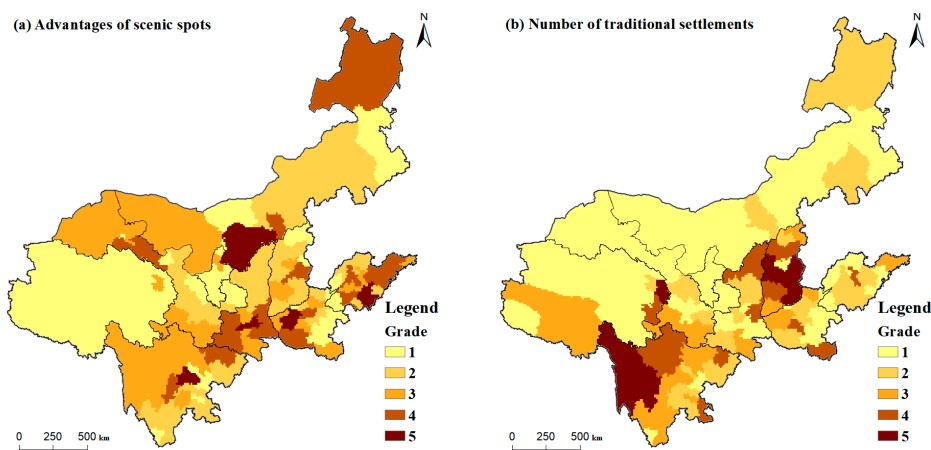

**Figure 6.** The evaluation grade of specific index of core resources.

(3)   External environment

Figure 7 gives the evaluation grade of the three specific indexes of the external environment. It can be seen that the external environment of the national ICH in the Yellow River Basin is obviously different. ① From the GDP per capita, there are few regions with high economic development levels (Grade 4 and above). There are 28 prefecture-level cities, accounting for 24% of the evaluation unit. In cities with a concentrated distribution of national ICH, the economic foundation is relatively weak. ② From the tourism income, the tourism development level of cities in the middle and lower reaches of the Yellow River is generally higher than that of cities in the upper reaches. There is a big gap in the spatial distribution of tourism development in the whole region. The cities with high grade evaluation only account for 10% of the evaluation unit. Among them, Chengdu, Xi'an, Qingdao, Zhengzhou, Luoyang, Jinan, Jinzhong, Taiyuan, Hohhot and other cities have outstanding tourism development advantages. From the traffic conditions, the areas with low grade evaluation (Grade 2 and below) account for 54% of the evaluation unit, and areas with high grade evaluation (Grade 4 and above) account for 17% of the evaluation unit. It is mainly distributed in some cities in the middle and lower reaches of the Yellow River, among which Chengdu in Sichuan province has the most prominent traffic advantages.

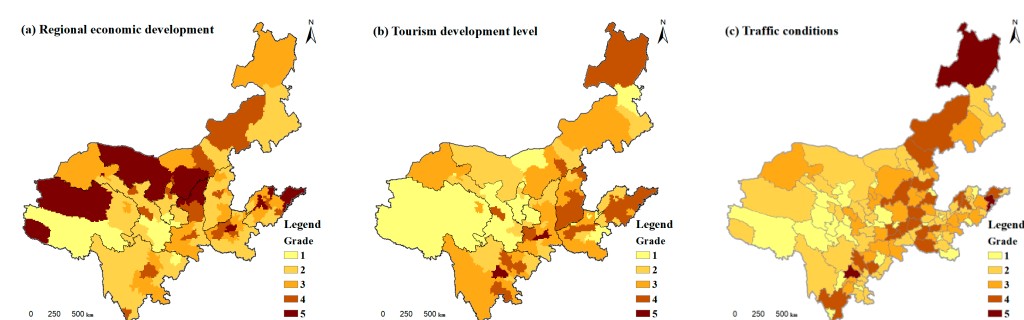

**Figure 7.** The evaluation grade of specific index of external environment.

4.2.3. Grading of Comprehensive Evaluation

Based on the spatial analysis function of ArcGIS10.8, the evaluation grades of eight specific indexes are weighted and analyzed by superposition to calculate the tourism utilization potential of national ICH in the Yellow River Basin. The natural breakpoint splitting method is used to divide the calculation results into five grades, and the grade map of comprehensive evaluation in the Yellow River Basin is then obtained (Figure 8). There are 72 cities with the evaluation grade of three or above, accounting for 63% of the

evaluation unit. It indicates that the tourism potential of national ICH in the Yellow River Basin is great.

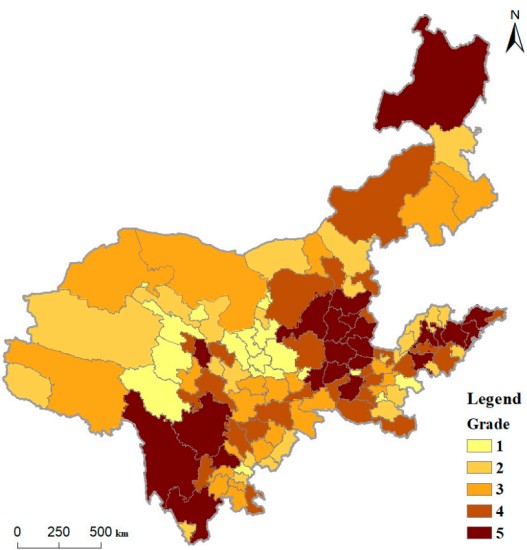

**Figure 8.** The grade of comprehensive evaluation.

(1) High potential area (Grade 4 and above). There are 49 cities in the area, accounting for 43% of the evaluation unit. Among them, there are 23 cities with highest grade evaluation (Grade 5), mainly located in the central and southern Shanxi province, the northern Shandong province, the cities in the central and western Sichuan province, Hulun Buir in Inner Mongolia, Haidong in Qinghai province, and Weinan in Shaanxi province. These cities are rich in national ICH resources. There are 26 cities with highest grade evaluation (Grade 5), all of which are located around the highest potential cities. Although these cities are supported by tangible carriers or external environmental conditions, they need to deeply tap their own characteristics of national ICH resources to develop different types of tourism.

(2) Medium potential area (Grade 3). There are 23 cities in the region, accounting for about 20% of the evaluation unit, and the distribution is relatively scattered. These cities have general advantages in national ICH resources, and their space carriers or external environmental conditions have only some advantages. If these cities can fully develop their advantageous resources, they still have certain tourism utilization potential.

(3) Low potential area (Grade 2 and below). There are 43 cities in the region, accounting for 37% of the evaluation units. These cities are mainly located in the middle and upper reaches of the Yellow River Basin, with no more than 10 national ICH. The tourism development level and economic development of these cities are relatively poor, which is related to their geographical location. In addition, there are many ethnic minorities in these cities, which are relatively isolated in all aspects, making it difficult to develop national ICH tourism.

## 5. Discussion

### 5.1. Discussion on Spatial-Temporal Pattern of National ICH Resources

In this study, the national ICH projects are considered as point geographical element. It is convenient for quantitative analysis of the spatial-temporal pattern of the national ICH in the Yellow River Basin. At present, the application of GIS technology in the research of ICHs is still lacking. Methods such as the nearest neighbor index, kernel density estimation, standard deviation ellipse and spatial autocorrelation method have been integrated in ArcGIS10.8, which are used to ensure that spatial-temporal pattern of national ICH resources in the Yellow River Basin are comprehensive and deeply explored.

Moreover, this study takes prefecture-level cities as evaluation units, and the research results are more precise.

The spatial distribution pattern of national ICH has strong spatial heterogeneity in the Yellow River Basin, whether from the overall perspective, the basin perspective (upstream, middle and downstream) or the provincial perspective. It is consistent with previous studies [53,54]. The national ICH resources in the Yellow River Basin show a spatial distribution pattern of "one belt and two cores", which is consistent with the research results on the spatial distribution characteristics for multicore spatial structures in other regions [48,50–54]. Moreover, the sub-high-density areas are located at the periphery of the high-density regional cities. The hot spots of national ICH in the Yellow River Basin are always located in most cities in central and southern Shanxi, Henan, and Shandong provinces, which is consistent with previous studies [48]. Xu and Pan noted that the high densities of ICH are concentrated in flat and water-rich regions. These areas have fertile soil, pleasant climate, long cultural history, ethnic agglomeration and development [48]. The cold spots are mainly concentrated in some cities in Inner Mongolia, Ningxia and Si-chuan provinces, which suggested that some measures of national ICH development could be conducted in these areas.

Existing studies pay more attention to the spatial distribution characteristics of ICH, but less on the temporal characteristics of ICH. Zhang et al. analyze the temporal change characteristics of ICH of the Yellow River Basin from the perspective of historical dynasties [54]. In this study, temporal change characteristics of national ICH distribution in the Yellow River Basin are analyzed according to the batch time published at the national levels. The nearest neighbor index, spatial distribution density, standard deviation ellipse, Moran's I index and hot spot analysis of five batches of national ICH are carried out separately in the Yellow River Basin. The five batches of national ICH in the Yellow River Basin present a temporal distribution pattern of "Northeast to Southwest". The distribution scope of national ICH in medium- and high-density areas expands gradually and finally tends to be balanced. This shows that China has gradually attached importance to the protection of national ICH resources in ethnic minorities and underdeveloped regions.

### 5.2. Discussion on the Tourism Utilization Potential of National ICH Resources

The development of national ICH tourism is an enabler of sustainable economic development and essential to the spiritual wellbeing of people for its powerful symbolic and aesthetic dimensions. China is promoting the Yellow River National Cultural Park Scheme to provide cultural welfare for the public. The Yellow River Basin will become the foothold to enhance the influence of river civilization in the new era. Therefore, the study of national ICH in the Yellow River Basin should not only pay attention to its spatial and temporal distribution characteristics but also further quantitatively assess its tourism utilization potential, so as to provide scientific reference for both better symbolizing heritage value within the region and making rational national ICH development strategies. However, there are few existing studies to comprehensively evaluate the utilization potential of national ICH tourism in the Yellow River Basin from a holistic and broad perspective.

In this study, the resource-market-product theory is used to establish an evaluation system of the tourism utilization potential of national ICH from three aspects: core resources, tangible carrier and external environment. There are 49, 23 and 43 cities in high-potential, medium-potential and low-potential areas, accounting for 43%, 20% and 37% of the evaluation unit, respectively. Therefore, it is necessary to implement the strategy of regional differentiated tourism development and regional linkage tourism development in the Yellow River Basin to tap the tourism utilization potential of national ICH resources.

### 5.2.1. Development Strategies of ICH Tourism Utilization Potential

(1) Regional differentiated development

For areas with high potential, where the economic, tourism and intangible cultural heritages resources are at a high level, a capacity-building initiative should be launched

to strengthen the innovative tangible display of national ICH resources and create core tourist attractions of national ICH by promoting the integration of culture and tourism. For example, Taiyuan, the capital city of Shanxi province, has superior external conditions, so it can develop national ICH creative industry parks [61]. There are many traditional settlements in the central and southern cities of Shanxi province, Haidong city of Qinghai province, Weinan of shaanxi province, Aba Tibetan and Qiang Autonomous Prefecture and Ganzi Tibetan Autonomous Prefecture of Sichuan province, and their brand effects can be used to develop ecological national ICH tourism of ancient village [50,61]. Shandong province can cooperate with cultural scenic spots to develop traditional handicrafts tourism. Modern digital technology can be used to display the production process and development history of handicrafts, so as to enhance the participation and interaction of tourists [52]. According to the construction plan of the Yellow River National Cultural Park, these regions should give priority to the development of high-quality national ICH tourism projects.

For areas with medium potential, it should give full play to their advantages and improve their disadvantages to develop national ICH tourism. For example, Xianyang city in Shaanxi province has excellent historical culture, profound cultural heritage and rich cultural relics. Although there are only four national ICHs, they can build the corresponding industrial base by relying on its convenient transportation conditions and significant tourism economic linkage development [50]. Under the construction planning of the Yellow River National Cultural Park, the tourism development of national ICH resources in these areas will also be supported.

For areas with low potential, the local government should strive to improve its economic strength and optimize the conditions of transportation facilities and infrastructure construction. By improving its comprehensive regional conditions, a higher level of tourism attraction can be created, which can effectively promote the tourism development of national ICH resources.

(2)   Regional linkage development

The tourism development of ICH resources in the Yellow River Basin should be based on sustainable development. The highlight of regional cultural characteristics should rely on the continuous promotion of cultural and tourism integration and high-quality development. The different national ICH tourism utilization potential areas in the Yellow River Basin also need to strengthen exchanges and cooperation, which can avoid repeated development and construction of tourism projects in the same region, and effectively improve the utilization efficiency of national ICH resources. For example, the Yellow River Basin can take the high-potential cities as the centers, and develop tourism development of national ICH resources by linking surrounding cities. By connecting the tourism routes of the same culture, it can form a concentric circle of diffusion.

So far, many national plans have been issued for the development of the Yellow River Basin, such as "The Belt and Road Initiative" and the "Development of Yellow River National Cultural Park". The Yellow River National Cultural Park is an important carrier for heritage protection in the new era and for promoting the high-quality development of cultural and tourism integration. Its core task is to comprehensively build the Yellow River cultural tourism brand and ensure that the iconic cultural heritage and tourist attractions will make a wonderful appearance [69]. Therefore, the development of national ICH tourism in different tourism utilization potential areas in the Yellow River Basin should also be combined with other plans. In the national key festivals, it is also suggested to further increase the elements of the Yellow River and attach importance to cultivating the brand of the region. The cultural and tourism corridors with international influence can be built by maximizing the characteristics of national ICH resources in the Yellow River Basin.

### 5.2.2. Strategies to Avoid the Commercialization of ICH Tourism

There has been a heated debate between the protection of ICH and the commercialization of tourism development, which is pronounced in developing countries [5]. Conservationists believe that ICH tourism gains benefits by compromising conservation

goals [4]. Markowska and Nowakowska pointed out that the development of martial arts tourism has greatly reduced the authenticity of ICH [6]. In fact, ICH tourism undoubtedly is a method to revitalize ICH with the largest social and economic benefits. Kim et.al explore the practitioner approach to the authenticity of ICH and ICH as a sustainable tourism resource in South Korea [9]. Ma and Liu pointed out that the integration of traditional technology and technology could maintain ecological sustainability, promote rural economic development and play a role in poverty alleviation in China [10].

In order to prevent the commodification of ICH from threatening its authenticity in the development of ICH tourism in the Yellow River Basin, this study also puts forward some strategies. The witness function of the continuous inheritance of the Yellow River civilization should be fully developed, and the basic premise of protection must be guaranteed. Then scientific protection rules, standards, methods and mechanisms should be formulated and implemented. It can avoid the risks brought by the commercialization of ICH tourism resources, adhere to the authenticity of ICH and ensure the inheritance, and protect the ICH. Meanwhile, the ICHs and their culture and natural ecological environment that can be nurtured and developed need whole protection.

## 6. Conclusions

The main conclusions of this study are as follows: (1) From the perspective of spatial distribution pattern, the national intangible cultural heritages in the Yellow River Basin are clustered, forming a "one belt and two cores" distribution pattern. The cities in Shanxi, Henan and Shandong provinces are high-density and sub-high-density continuous belts, and Haidong city and Chengdu city are the high-density and sub-high-density core areas, respectively. The five batches of national intangible cultural heritages resources present a temporal distribution pattern of "Northeast to Southwest" and finally tend to be distributed in a balanced mode. Most cities in central and southern Shanxi, Henan and Shandong provinces are always hot spots. (2) From the perspective of the evaluation grade of tourism utilization potential, there is an obvious gap in the evaluation grade of specific indexes among cities in the Yellow River Basin. However, for the evaluation grade of comprehensive evaluation, there are 72 cities with a grade of three or above, accounting for 63%. The results indicate that the tourism potential of national intangible cultural heritages in the Yellow River Basin is great. (3) The tourism utilization potential of national intangible cultural heritages in the Yellow River Basin can be exploited from two aspects: regional differentiation development and regional linkage development to coordinate and promote the high-quality development of tourism in the region. These development strategies can effectively promote the integrated development of national ICH resources and tourism, promote the inheritance and protection of the national ICH, and coordinate the high-quality development of tourism in the Yellow River Basin.

In the future, we can analyze the spatial distribution pattern of various types of national intangible cultural heritages in prefecture-level cities of the Yellow River Basin. Due to the poor availability of some data, the indexes selected may not be comprehensive enough when building the evaluation index system of tourism utilization potential of national intangible cultural heritage resources. Moreover, the applicability of the development strategy of national intangible cultural heritage tourism proposed in this study in other regions still needs to be verified.

Intangible cultural heritage can effectively promote sustainable development from all dimensions. Therefore, the protection of intangible cultural heritage is indispensable if communities around the world want to realize the dreamed future. The development of ICH tourism can not only promote the sustainable development of the regional economy but also strengthen the protection of ICH resources. With the need for high-quality development of the Yellow River Basin, the national intangible cultural heritage tourism in the region will be more systematic and diversified. Therefore, exploring a sustainable and balanced development model of ICH is one of the major future research objectives.

**Author Contributions:** Conceptualization, B.C. and R.Z.; data collection, B.C. and X.L.; methodology, B.C.; data analysis, B.C.; writing—original draft preparation, B.C.; writing—review and editing, B.C. and X.D.; supervision, X.D and J.X. All authors have read and agreed to the published version of the manuscript.

**Funding:** This research was funded by Scientific Research Program of Tianjin Municipal Education Commission (2017SK087), The Second Comprehensive Scientific Expedition on the Qinghai Tibet Plateau (2019QZKK1004), Innovation Team Training Program of universities in Tianjin during the 13th Five Year Plan Period (TD13-5093), Scientific Research Program of Tianjin Municipal Education Commission (2019SK021), National Natural Science Foundation of China (41671156).

**Institutional Review Board Statement:** Not applicable.

**Informed Consent Statement:** Not applicable.

**Data Availability Statement:** All data generated or analyzed during this study are included in this published article.

**Acknowledgments:** We acknowledge the Chinese Ministry of Culture and Tourism for providing national intangible cultural heritage data. We also acknowledge the provincial people's government websites in China, the Chinese government website, the national traditional village website and provincial statistics bureau websites in China for providing the evaluation index data of tourism utilization potential. The figures are generated using the ArcGIS10.8 software.

**Conflicts of Interest:** The authors declare no conflict of interest.

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
