# Peer review of "Spatial-Temporal Distribution Pattern and Tourism Utilization Potential of Intangible Cultural Heritage Resources in the Yellow River Basin"

_sustainability, doi:10.3390/su15032611_

Round 1

Reviewer 1 Report

Dear Authors,

The manuscript presents the results of interesting studies. In my opinion, slight adjustments can be made.

Remarks:

1.     The abstract should be a total of about 200 words maximum. It is much longer in this manuscript.

2.     No research hypotheses / research questions were formulated. In addition, the methodological section is described correctly.

3.     The section containing the conclusions lacks the link between the research results and the theory presented in the literature review. I think this section should be expanded.

4.     I suggest emphasizing the relationship of the presented research with the subject of the journal. This is especially missing in section ‘6. Conclusions and Limitations’.

5.     There is no DOI in the references

Author Response

Dear Reviewer,

We are indebted to you for your insightful comments and suggestions. We have studied the valuable comments from you carefully, and tried our best to revise the manuscript. We hereby provide a summary of point-to-point responses to each of the comments. Suggested changes are incorporated into the revised manuscript in red text. Please see the attachment.

Kind regards,

All authors of this manuscript

Reviewer 2 Report

It is an interesting work, well organized and presented..

Some comments that could improve the specific research work..

1.     The methodology followed in the paper, should be accompanied by relevant references. Οther research work that use the specific or similar to that, process. There should be a justification for the use of the specific methodology. Is it more appropriate for such an analysis? Is it better than others and it can provide better results? or it has been used in similar analysis for other places? etc... In the presentation of the methodology, only one reference exist, relative to autocorrelation.

2.     Section 5 includes some guidelines for improvements in touristic activity in the areas analyzed. However, they seem to be only authors’ propositions. Some arguments that may strengthen those propositions should be provided and will be related to the previous analysis.

 3.     The discussion section should relate the findings with some prepositions, connecting the theoretical background (literature) with the findings and proceed to specific policy measures and so on. That is not clear from the paper. ..... 

In table 4, the sum of the weights shouldn't be 1 (or 100)? The sum is 1,063 (or 106,3). Furthermore, some justification for those indexes (or some references of similar use in other works) could improve the presentation of the analysis.

Author Response

Dear Reviewer,

We are indebted to you for your insightful comments and suggestions. We have studied the valuable comments from you carefully, and tried our best to revise the manuscript. English language and style have been spell check. We hereby provide a summary of point-to-point responses to each of the comments. Suggested changes are incorporated into the revised manuscript in red text. Please see the attachment.

Kind regards,

All authors of this manuscript

Reviewer 3 Report

It is a study that does not substantiate the fundamental issues that justify the need for the study. Above all, there is little criticism of the risks of commodification of the ICH as a tourism driver, which is taken for granted when there is much controversy on this issue. Also in line with the above-mentioned, the weak link to the field of sustainability is highlighted, the paper should reinforce the contribution of this study to this field as it is not clear. It is also worth noting the ambiguity in the use of concepts that need to be clarified regarding the approach taken. This is the case of concepts such as protection or authenticity.
Also, throughout the article there is information that is not properly referenced.
These issues are of fundamental importance in any research, which is why a more in-depth study is recommended, even if the quantitative work has been carried out correctly.

Author Response

(The authors gave the same response as above.)

Round 2

Reviewer 1 Report

Thank you very much for making improvements in accordance with my suggestions and providing justifications. Unfortunately, I cannot agree with one. The scientific article presents the results of scientific research. Research is conducted to verify a research hypothesis(s) or to answer a research question(s). I agree that research hypotheses are not always formulated. But in such a situation, the research results should answer the research questions that are the basis of the research process. Therefore, I believe that this is missing from this manuscript.

Reviewer 3 Report

 The authors have tried to give an answer to the main highlighted issues in the previous report, but unfortunately, it seems not to have been enough. The key points are that the study's primary justification seems weak and continues without giving a strong answer to the main associated risks. There is little criticism of the dangers of the commodification of the ICH as a tourism driver, which is taken for granted when there is much controversy on this issue. Furthermore, the ambiguity in the use of concepts that need to be clarified regarding the approach taken hasn't been fixed. This is the case with concepts such as protection or authenticity.

Also, the discussion should incorporate this point in the section.

These issues should be fixed in the introduction section and / or the theoretical framework before the paper can be recommended for publishing.

Round 3

Reviewer 1 Report

Thank you. My decision: Accept in present form

Author Response

Dear Reviewer,

We are indebted to your insightful comments and suggestions again. And the quality of our manuscript has been greatly improved.

Kind regards,

All authors of this manuscript

Reviewer 3 Report

The authors have answered the main associated risks. They incorporated reflections about the risks of the commodification of the ICH as a tourism driver, but it continues to be missing in the discussion section.
